# GPC1 specific CAR-T cells eradicate established solid tumor without adverse effects and synergize with anti-PD-1 Ab

Daiki Kato[1,2], Tomonori Yaguchi[1]*, Takashi Iwata[1,3], Yuki Katoh[1], Kenji Morii[1], Kinya Tsubota[1,4], Yoshiaki Takise[1], Masaki Tamiya[1], Haruhiko Kamada[5], Hiroki Akiba[5], Kouhei Tsumoto[5], Satoshi Serada[6], Tetsuji Naka[6], Ryohei Nishimura[2], Takayuki Nakagawa[2], Yutaka Kawakami[1,7]*

[1]Division of Cellular Signaling, Institute for Advanced Medical Research, Keio University School of Medicine, Tokyo, Japan; [2]Laboratory of Veterinary Surgery, Graduate school of agricultural and life sciences, The University of Tokyo, Tokyo, Japan; [3]Department of Obstetrics and Gynecology, Keio University School of Medicine, Tokyo, Japan; [4]Department of Ophthalmology, Tokyo Medical University, Tokyo, Japan; [5]Center for Drug Design Research, National Institute of Biomedical Innovation, Health and Nutrition, Tokyo, Japan; [6]Center for Intractable Immune Disease, Kochi Medical School, Kochi University, Kochi, Japan; [7]Department of immunology, School of Medicine, International University of Health and Welfare, Tokyo, Japan

**Abstract** Current xenogeneic mouse models cannot evaluate on-target off-tumor adverse effect, hindering the development of chimeric antigen receptor (CAR) T cell therapies for solid tumors, due to limited human/mouse cross-reactivity of antibodies used in CAR and sever graft-versus-host disease induced by administered human T cells. We have evaluated safety and antitumor efficacy of CAR-T cells targeting glypican-1 (GPC1) overexpressed in various solid tumors. GPC1-specific human and murine CAR-T cells generated from our original anti-human/mouse GPC1 antibody showed strong antitumor effects in xenogeneic and syngeneic mouse models, respectively. Importantly, the murine CAR-T cells enhanced endogenous T cell responses against a non-GPC1 tumor antigen through the mechanism of antigen-spreading and showed synergistic antitumor effects with anti-PD-1 antibody without any adverse effects in syngeneic models. Our study shows the potential of GPC1 as a CAR-T cell target for solid tumors and the importance of syngeneic and xenogeneic models for evaluating their safety and efficacy.

*For correspondence:
beatless@rr.iij4u.or.jp (TY);
yutakawa@keio.ac.jp;
yutakawa@iuhw.ac.jp (YK)

**Competing interests:** The authors declare that no competing interests exist.

## Introduction

Immunotherapies with a chimeric antigen receptor T (CAR-T) cells have been demonstrated robust clinical responses in hematological malignancies. However, developments of CAR-T cell therapies for solid tumors have met several obstacles including on-target off-tumor lethal toxicities, insufficient activation of administered CAR-T cells in tumor tissues, and loss of target antigens in tumor cells (*June et al., 2018*; *Kato et al., 2017*). For example, even weak expression of target antigen on normal tissues caused lethal toxicities in the first-in-human clinical trial (*Morgan et al., 2010*). To develop CAR-T cell therapies for solid tumors, it is important to identify suitable target antigens and to evaluate their safety as well as their antitumor efficacy in the preclinical models.

Majority of preclinical studies for CAR-T cell therapies have been performed in vivo using xenogeneic mouse models wherein response of human T cells against human xenografted tumors are tested

(*Li et al., 2017*; *Shiina et al., 2016*; *Carpenito et al., 2009*). This strategy is often the only available option for development of CAR-T cells therapies targeting human tumors, due to the lack of cross-reactivity of CAR against the homologous mouse antigen. However, because of the restricted CAR cross-reactivity, the effects of human CAR-T cells against normal tissues cannot be fully studied in xenogeneic models. Indeed, preclinical studies of CAR-T cell therapies in xenogeneic mouse model could not predict the lethal toxicities that occurred in the first-in-human clinical trial of HER2-specific CAR-T cells (*Morgan et al., 2010*; *Zhao et al., 2009*). On the other hand, if the human CAR-T cells can recognize mouse homogenous antigens, syngeneic mouse models can be used to evaluate the on-target off-tumor toxicities (*Siegler and Wang, 2018*). Moreover, such syngeneic models enable analyses of the effects of CAR-T cell therapy on endogenous antitumor immune responses and efficacy of combined therapies that are limited in xenogeneic mouse models due to their incompetent host immunity and graft-versus-host disease (GVHD) by the xenografted T cells (*Yaguchi et al., 2018*).

Glypican-1 (GPC1) is a cell-surface heparansulphate proteoglycan expressed in normal fetal tissues and tumor cells. A major function of GPC1 is its involvement in the development of brain in fetal phase. While knockout of GPC1 exhibits no abnormalities in morphology, behavior, or life span in adult mice, reduction in brain volume has been reported in early fetal phase (*Jen et al., 2009*). Thus, GPC1 seems to have no critical function in healthy adult stage. Recently, we and other groups have reported the overexpression of GPC1 in various human cancers including glioma, mesothelioma, several squamous cell carcinomas (SCC) such as esophageal and cervical cancers, and several adenocarcinomas such as breast and pancreatic cancer (*Hara et al., 2016*; *Matsuzaki et al., 2018*; *Matsuda et al., 2001*; *Melo et al., 2015*; *Duan et al., 2013*; *Su et al., 2006*; *Amatya et al., 2018*). GPC1 expression in cancer is linked with malignant phenotypes such as promotion of cell-cycle and enhanced metastatic potential, and is considered as a prognostic factor in some cancers (*Kleeff et al., 1998*). In order to develop a cancer therapy targeting GPC1, we have previously generated anti-GPC1 cytotoxic monoclonal antibodies (mAb) (clone: 1–12). This anti-GPC1 mAb recognized both human and mouse GPC1, and inhibited the growth of GPC1-expressing human esophageal SCC xenografted in immunodeficient mice without any obvious adverse effects (*Harada et al., 2017*).

In this study, we generated GPC1-specific CAR-T cells from the variable regions of the anti-GPC1 mAb (clone: 1–12) that recognize both human and mouse GPC1, and evaluated their efficacy and safety, using not only tumor xenografts in immunodeficient mouse models but also immunocompetent syngeneic mouse models.

## Results

### Low protein expression of hGPC1 was detected by anti-GPC1 mAb (clone: 1–12) in normal human adult tissues

We and others have reported that human GPC1 (hGPC1) was preferentially expressed in various tumors and fetal brain (*Jen et al., 2009*; *Hara et al., 2016*). To confirm tumor specific expression of hGPC1, we first analyzed expression of *hGPC1* mRNA in cervical squamous cell carcinoma tissues, various adult human normal tissues, and fetal brain tissues by qPCR analysis. Most of the cervical squamous cell carcinoma tissues expressed higher *hGPC1* mRNA than corresponding normal cervix tissues and various normal tissues (*Figure 1A and B*).

Next, we evaluated the reactivity of anti-GPC1 mAb (clone: 1–12) against various human normal tissues by immunohistochemistry (IHC). Compared to its high expression in human esophageal carcinoma, normal tissues showed low to no expression of GPC1 when stained with anti-GPC1 mAb (clone: 1–12). We confirmed this finding in tissue samples from three donors of different age and sex, and representative data is shown in *Figure 1C*. These data indicated that GPC1 would be a promising therapeutic target for CAR-T cell therapies and anti-GPC1 mAb (clone: 1–12) could be used for the generation of CAR-T cells.

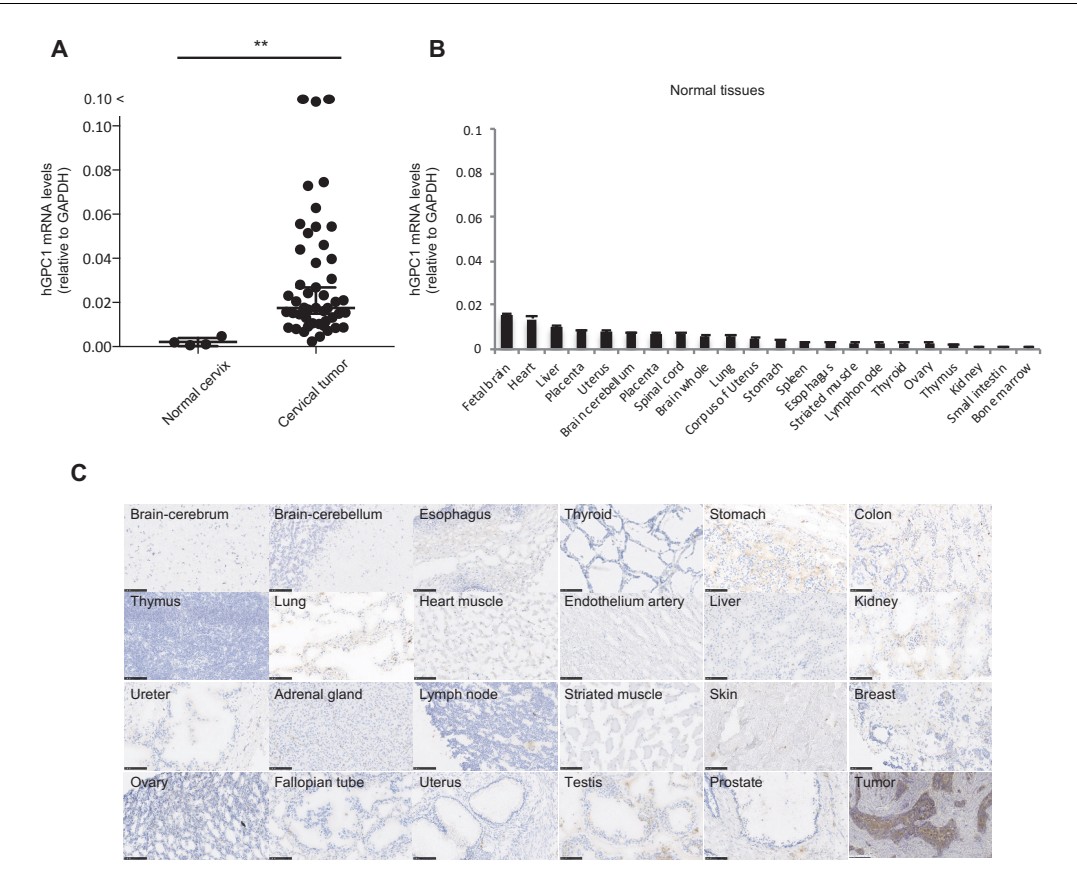

**Figure 1.** Low protein expression of GPC1 in human normal tissues detected by anti-GPC1 mAb (clone: 1–12). (**A**) The mRNA expression of hGPC1 was evaluated by qPCR in human normal cervix and cervical squamous carcinoma tissues; GAPDH was used as an internal control. (**B**) The mRNA expression of hGPC1 was evaluated by qPCR in various human adult normal tissues and human fetal brain tissue; GAPDH was used as an internal control. (**C**) IHC staining by anti-GPC1 mAb (clone: 1–12) in human adult normal tissues and human esophageal SCC tissues. Scale bar, 100 μm.

## hCAR-T cells derived from the scFv of anti-GPC1 mAb (clone: 1–12) specifically recognized hGPC1-positive tumor cells and targeted xenografted solid tumors in vivo

In order to generate GPC1-specific hCAR, VH and VL chains of anti-GPC1 mAb (clone: 1–12) were used for scFv fragment of the CAR. Surface plasmon resonance (SPR) analysis showed high binding affinity of LH or HL forms of scFv against recombinant hGPC1 protein as calculated $K_D$ value 9.06 × $10^{-9}$ M or 1.22 × $10^{-8}$ M, respectively, which was as high as that of anti-CD19 scFv currently used in clinical settings (*Ghorashian et al., 2019*). The generated scFv was then connected to the signal domains of human CD28 and CD3ζ and made into retroviral expression vector for transduction into activated human T cells (*Figure 2A* and *Figure 2—figure supplement 1*). There were no significant differences between LH form and HL form of hCAR-T cells in their proliferations after transfection (data not shown).

Since cytokine secretion and killing activity of T cells in response to target antigen is important in the antigen-specific antitumor immune response, we tested these abilities of the hCAR-T cells by IFNγ-releasing assay and $Cr^{51}$ releasing assay. When GPC1-specific hCAR-T cells were co-cultured with LK2-hGPC1 (a hGPC1 overexpressing lung carcinoma cell line), TE14 (an esophageal carcinoma cell line with endogenous hGPC1), or LK2-mock (a hGPC1-negative lung carcinoma cell line), IFNγ secretion and $Cr^{51}$ release were detected in hGPC1-dependent manner (*Figure 2B,C and D*). Our results also indicated that the HL form of CAR possessed stronger antitumor activity than the LH form.

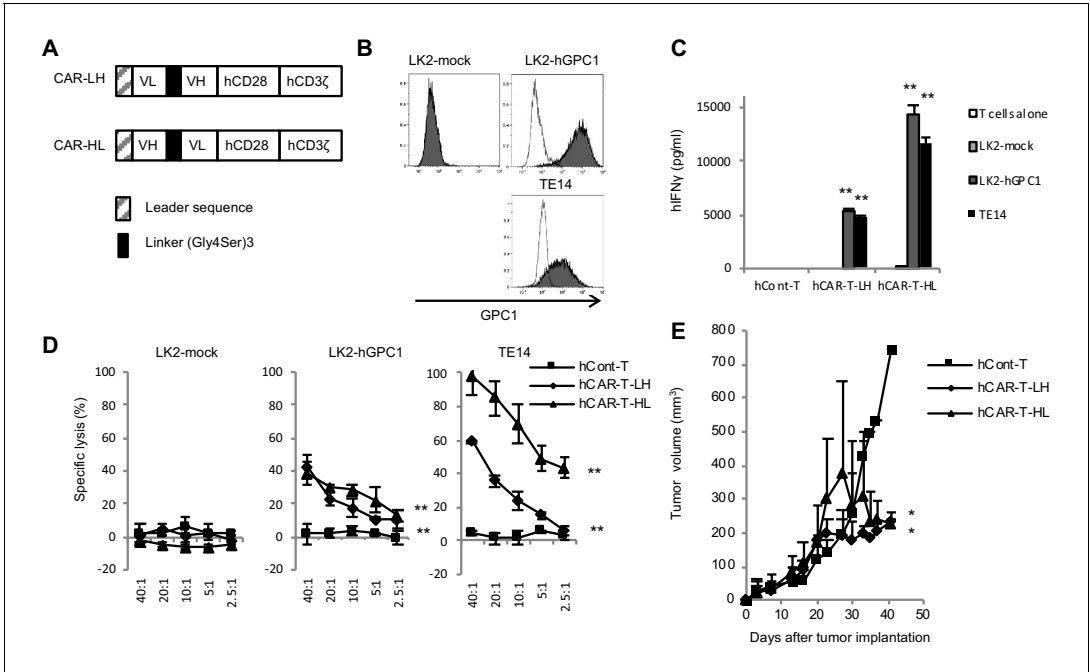

**Figure 2.** GPC1-specific human hCAR-T cells specifically recognized hGPC1-positive tumor cells and inhibited tumor growth in xenograft mouse model. (A) Diagrams of GPC1-specific human hCAR; scFv frgments derived from light chain (VL) and heavy chain (VH) of anti-GPC1 mAb (clone: 1–12) were fused to human CD28 and human CD3ζ signal domains. The positions of VL and VH were switched to generate two forms of CAR gene, LH and HL. (B) LK2-hGPC1, LK2-mock, and endogenous hGPC1-expressing TE14 were stained by anti-GPC1 mAb (clone: 1–12) (shaded histogram) or isotype control (open histogram). (C) GPC1-specific IFNγ secretion of hCAR-T cells (LH or HL form) or hCont-T cells co-cultured with LK2-mock, LK2-hGPC1, or TE14. (D) Antigen-specific in vitro cytotoxicity of hCAR-T cells (LH or HL form) or hCont-T cells against LK2-hGPC1, LK2-mock, or TE14 was evaluated by using standard $Cr^{51}$ releasing assay. (E) hCAR-T cells (LH or HL form) or hCont-T cells ($2 \times 10^7$ cells/mouse) were injected into TE14-bearing NOG mice on day 9. Results are representative of two or three experiments. Error bars indicate SD.

The online version of this article includes the following source data and figure supplement(s) for figure 2:

**Figure supplement 1.** The sequences of GPC1-specific human CAR vectors.

**Figure supplement 1—source data 1.** The sequences of GPC1-specific human CAR vectors.

We further explored antitumor activities of the GPC1-specific hCAR-T cells in vivo. The hCAR-T cells were intravenously administered in TE14 xenografted NOG mice. The hCAR-T cells effectively inhibited tumor growth, compared to hCont-T cells (*Figure 2E*). These results indicated specific recognition and strong antitumor efficacy of the GPC1-specific hCAR-T cells against human tumors expressing hGPC1.

## Low protein expression of mGPC1 was detected by anti-GPC1 mAb (clone: 1–12) in normal murine adult tissues

Due to administration of xenogeneic T cells, all NOG mice treated with hCAR-T cells suffered from and died of severe GVHD. Consequently, it was difficult to evaluate on-target off-tumor adverse effect in xenogeneic model. Thus, we established a syngeneic mouse model for the evaluation of on-target off-tumor adverse effects. First, we evaluated the expression profile of *mGpc1* in normal tissues of adult mouse by qPCR, and detected low levels of mRNA expression in various tissues. Similar to human tissues, heart, lung, and brain of mouse showed relatively high expression of *Gpc1* mRNA (*Figures 1A* and *3A*).

Next, we evaluated reactivity of anti-GPC1 mAb (clone: 1–12) to various normal tissues of mouse by IHC. Similar to the observation in human adult tissues, anti-GPC1 mAb (clone: 1–12) detected very low to no expression of GPC1 in normal tissues of adult mouse (*Figure 1C* and *Figure 3B*). These results indicated similar expression pattern of GPC1 between human and mice at mRNA and

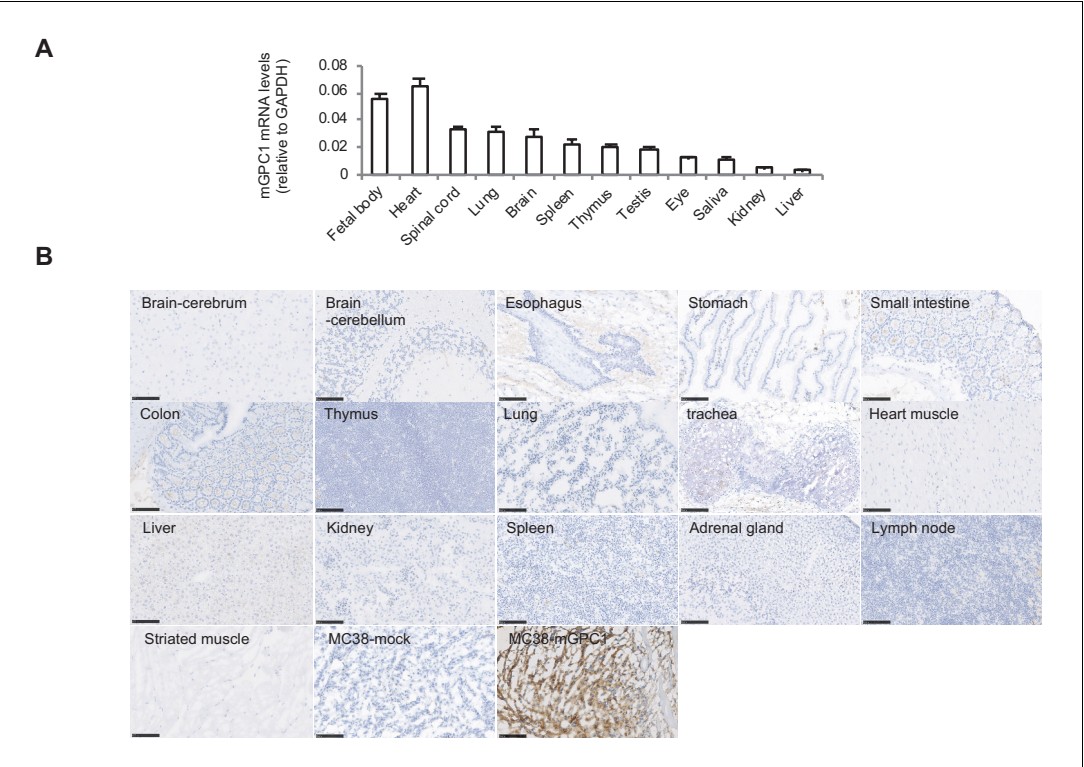

**Figure 3.** Low protein expression of GPC1 in murine normal tissues detected by anti-GPC1 mAb (clone: 1–12). (**A**) Expression of mGPC1 mRNA in various mouse normal tissues and GPC1-positive fetal bodies was quantified by qPCR; GAPDH was used as an internal control. (**B**) The mouse normal tissues and GPC1-positive MC38-mGPC1 tissues were IHC stained by anti-GPC1 mAb (clone: 1–12) in. Scale bar, 100 μm.

protein levels and suggested that syngeneic mouse model (C57BL/6) could be used for the evaluation of on-target off-tumor adverse effects.

## mGPC1-specific murine CAR (mCAR)-T cells specifically recognized mGPC1 and eradicated established mGPC1 expressing solid tumors

To generate GPC1-specific CAR-T cells from murine T cells and evaluate antitumor effects and adverse effects on normal tissues in syngeneic mouse models, we evaluated binding affinity of both LH and HL forms of scFv against recombinant mGPC1 protein by SPR analysis, and found that both LH ($K_D$ 5.18 × 10$^{-8}$ M) and HL ($K_D$ 1.09 × 10$^{-7}$ M) forms of scFv showed high binding affinity against mGPC1, although less affinity than those against hGPC1. The HL form was used for generating GPC1-specific mCAR-T cells, because the HL form of hCAR-T cells showed higher antitumor activity in vitro (*Figure 2C and D*). All the human sequences in the hCAR vector (HL form) were converted to homologous murine sequences (*Figure 4A* and *Figure 4—figure supplement 1*). The murine CAR vectors were efficiently transduced to murine activated T cells derived from the splenocytes of GFP transgenic mice that ubiquitously express GFP. Because we could not find mouse tumor cell lines which endogenously express mGPC1, we generated the mGPC1 overexpressing mouse tumor cell lines (MC38-mGPC1 and MCA205-mGPC1) which showed similar expression levels of mGPC1 compared to LK2-hGPC1 and TE14 (*Figure 4B*). Cytokine secretion assay and cytotoxic assay were performed by co-culturing mCAR-T cells with MC38-mGPC1 in vitro. In these assays, we observed antigen-specific cytokine secretion and killing activity of GPC1-specific mCAR-T cells (*Figure 4C and D*).

To explore antitumor activity and on-target off-tumor adverse effects of mCAR-T cells in vivo, mCAR-T cells were intravenously injected into mice bearing MC38-mGPC1 or MCA205-mGPC1. Robust antitumor effects by mCAR-T cells against the established solid tumors were observed

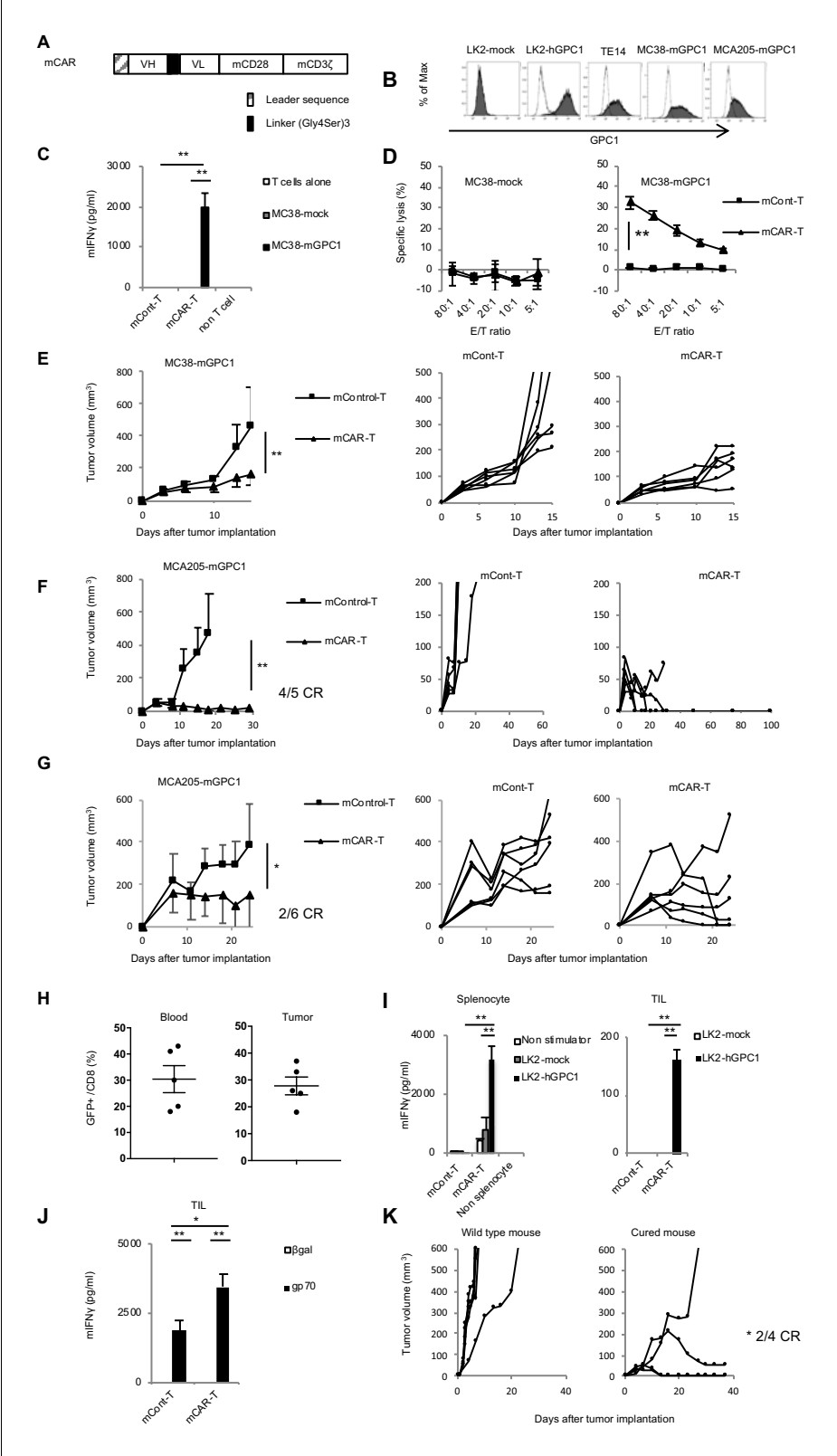

**Figure 4.** GPC1-specific murine mCAR-T cells specifically recognized mGPC1-positive tumor cells and eradicated solid tumors in vivo. (A) Diagram of GPC1-specific murine CAR; scFv fragment (HL) derived from anti-GPC1 mAb (clone: 1–12) was fused to mouse CD28 and CD3ζ signal domains. (B) The mGPC1-overexpressing murine cells (MC38-mGPC1 and MCA205-mGPC1), endogenous hGPC1-expressing human cells (TE14), and hGPC1-overexpressing human cells (LK2-hGPC1), hGPC1-negative cells (LK2-mock) were stained with anti-GPC1 mAb (shaded histogram) or isotype control

*Figure 4 continued on next page*

Figure 4 continued

(open histogram). (C) Antigen-specific IFNγ secretion of mCAR-T cells or mCont-T cells co-cultured with MC38-mGPC1 or MC38-mock was evaluated. (D) Antigen-specific cytotoxicity of mCAR-T cells or mCont-T cells against MC38-mGPC1 and MC38-mock was evaluated by using standard Cr$^{51}$ releasing assay. (E and F) Mice bearing MC38-mGPC1 tumor (E) or MCA205-mGPC1 tumor (F) received $2 \times 10^6$ cells of mCAR-T cells or mCont-T cells on day 3. Mean tumor volumes (mm$^3$ ± SD) of each group (left panels) and tumor-growth curves of the individual mice in each group (right panels) are shown. (G) Mice bearing MCA205-mGPC1 large tumor (tumor volume is >100 mm$^3$) received $3.5 \times 10^7$ cells of mCAR-T cells or mCont-T cells on day 7. Mean tumor volumes (mm$^3$ ± SD) of each group (left panels) and tumor-growth curves of the individual mice in each group (right panels) are shown. (H) Percentages of GFP-positive CD8$^+$ mCAR-T cells in total CD8$^+$ T cells from peripheral blood and tumor tissues on day 15 are shown. Dots indicate mice in each group. (I) Splenocytes (left panel) or CD8$^+$ TIL (right panel) were collected from the mice treated with mCAR-T cells or mCont-T cells on day 15, and co-cultured with LK2-hGPC1 or LK2-mock. After 24 hr, IFNγ in the supernatants was measured by ELISA. (J) CD8$^+$ TIL collected from the mice treated with mCAR-T cells or mCont-T cells were re-stimulated with irradiated normal splenocytes pulsed with gp70 peptides. After 48 hr, the re-stimulated TIL were collected and co-cultured with murine tumor cells pulsed with gp70 or control peptide (βgal) for 24 hr and IFNγ in the supernatants was measured by ELISA. Results are representative of two or three experiments. Error bars indicate SD. (K) 120 days after the mCAR-T cell administration, mGPC1-negative parental MCA205 was inoculated in the naive mice with no history of bearing tumors or the mice which had rejected MCA205-mGPC1 by mCAR-T cells injection. Tumor-growth curves of the individual mice in each group are shown.

The online version of this article includes the following source data and figure supplement(s) for figure 4:

**Figure supplement 1.** The sequences of GPC1-specific murine CAR vectors.
**Figure supplement 1—source data 1.** The sequences of GPC1-specific murine CAR vectors.

---

without any obvious adverse effects. In the MC38-mGPC1 mouse model, mCAR-T cells were effective in inhibiting tumor growth compared to mCont-T cells (*Figure 4E*). In the MCA205-mGPC1 mouse model, four out of the five mice receiving the mCAR-T cells showed complete tumor eradication lasting at least 100 days (*Figure 4F*). The mCAR-T cells were also injected into mice bearing large MCA205-mGPC1 tumor (tumor volume >100 mm$^3$) and significant in vivo antitumor activity was observed without any obvious adverse effects (*Figure 4G*). Further, we confirmed that the injected mCAR-T cells persisted in the peripheral blood and infiltrated into the tumor in MC38-mGPC1-bearing mice 15 days after the administration (*Figure 4H*). In the MCA205-mGPC1 mice that resulted in complete eradication of the tumors, we found that the mCAR-T cells were present in the peripheral blood for over 60 days (data not shown). To evaluate the functional persistence of the injected mCAR-T cells, cytokine secretion was evaluated ex vivo. 15 days after the mCAR-T cell injection, CD8$^+$ T cells were collected from the spleens and tumor tissues of MC38-mGPC1-bearing mice and co-cultured with LK2-hGPC1 or LK2-mock. The results indicated that the mCAR-T cells in both circulation and tumor retained GPC1-specific IFNγ producing activity for at least 15 days (*Figure 4I*).

Next, we sought to determine whether CAR-T cells targeting a single antigen could enhance T cell responses against other endogenous antigens. For this, CD8$^+$ TIL were harvested from the tumor of MC38-mGPC1-bearing mice and re-stimulated in vitro with gp70, an epitope peptide of MC38 for induction of gp70-specific T cell response. As shown in *Figure 4J*, gp70-specific IFNγ production was significantly enhanced by the administration of mCAR-T cells. Moreover, to evaluate the presence and importance of endogenous T cell responses in vivo, mGPC1-negative parental MCA205 tumor cells were re-challenged into the mice which had complete tumor eradication by mCAR-T cell therapy in the MCA205-mGPC1 model (*Figure 4F*). These mice showed a significant growth inhibition of the inoculated mGPC1-negative tumors, indicating the induction and involvement of T cells specific for endogenous tumor antigens in the strong antitumor effects of GPC1-specific CAR-T cell therapy in the syngeneic tumor modes (*Figure 4K*). The results from the syngeneic mouse models suggested that mCAR-T cells could eliminate established solid tumors and display persistent antitumor activity without obvious adverse effects. Furthermore, GPC1-specific CAR-T cell therapy enhanced antitumor responses of TIL against non-GPC1 endogenous tumor antigens.

## Adverse effects were not observed in normal tissues of mice treated with GPC1-specific mCAR-T in vivo

To evaluate adverse effects of mCAR-T cells in vivo, clinical symptoms and histological abnormalities were analyzed. No significant difference was observed in body weights of mice treated with either mCAR-T cells or mCont-T cells (*Figure 5A*). Similarly, no differences of gross appearance or behavior were observed between the treated and control mice. Histologically, no obvious tissue damage or infiltration of mCAR-T cells was observed in any of tested organs including the heart and brain that

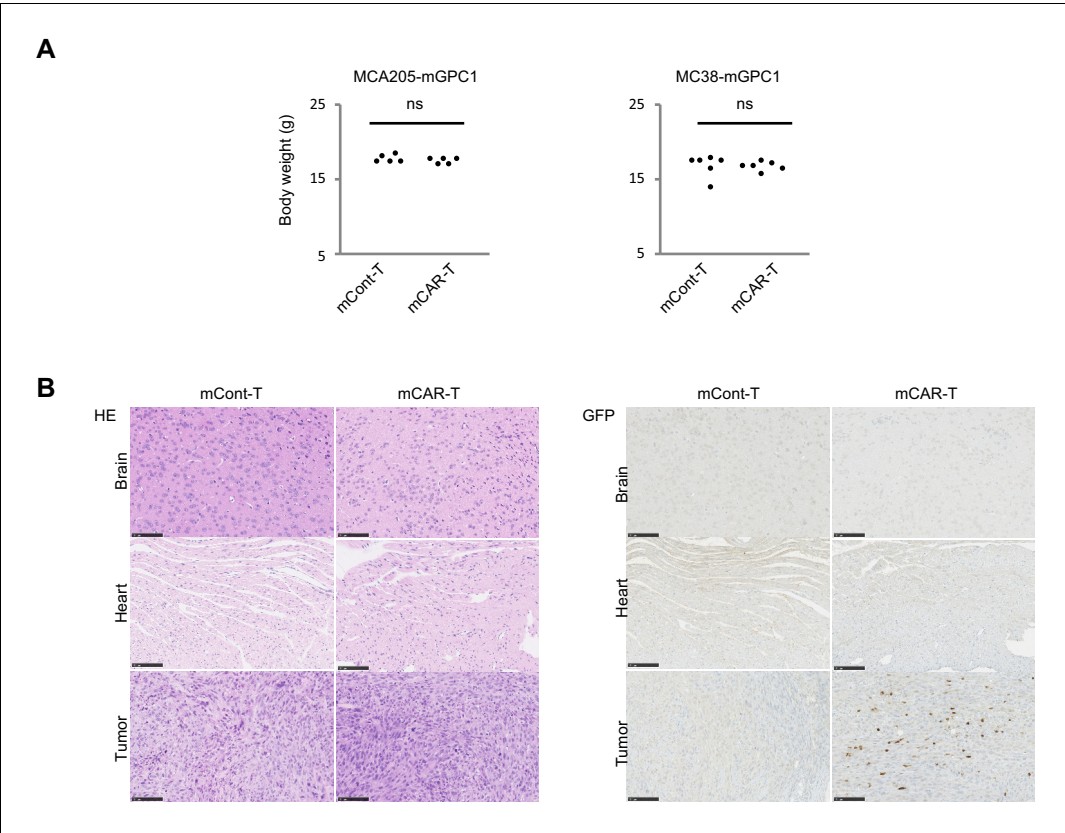

**Figure 5.** GPC1-specific mCAR-T cells showed no obvious adverse effects on normal tissues in vivo. (**A**) Body weight of mCAR-T cells or mCont-T cells injected mice bearing MC38-mGPC1 (right panel) or MCA205-GPC1 (left panel) was measured 12 days after mCAR-T cell or mCont-T cell administration. (**B**) Representative staining of HE (left panel) and IHC for injected GFP⁺ T cells detected by anti-GFP Ab (right panel) in mouse normal tissues are shown. The data of other normal tissues are shown in Figure 5-figure supplements 1 and 2.
The online version of this article includes the following figure supplement(s) for figure 5:

**Figure supplement 1.** Tissues damages were not detected in mouse normal tissues.
**Figure supplement 2.** Only few GFP-positive mCAR-T cells infiltrated mouse normal tissues.

showed detectable expression of GPC1 mRNA by qPCR (*Figure 5B* and *Figure 5—figure supplements 1* and *2*). Furthermore, administration of more than 10 times higher numbers of CAR-T cells did not show obvious adverse effects (data not shown) with significant antitumor effects against larger tumor (*Figure 4G*). These results suggested that mCAR-T cells could eliminate mGPC1-expressing tumors without any obvious adverse effects.

## GPC1-specific mCAR-T cells show synergistic antitumor activity in combination with anti-PD-1 antibody therapy

In MC38-mGPC1 mouse model, most of mCAR-T cells and endogenous T cells in tumor tissues expressed PD-1 (*Figure 6A*). The frequency of PD-1 expression on mCAR-T cells tended to be higher than that on mCont-T cells, although it did not show significance. As MC38-mGPC1 stably expressed PD-L1, we assessed whether anti-PD-1 Ab could enhance antitumor activities of the mCAR-T cell therapy (*Figure 6B*). The combination therapy of mCAR-T cells and anti-PD-1 Ab resulted in stronger antitumor activity than that of the mCAR-T cells alone (*Figure 6C*). Antitumor effects of anti-PD-1 Ab were relatively weak due to preconditioning with total body irradiation for in vivo CAR-T cell expansion. While the mice treated with either anti-PD1 Ab or mCAR-T cells alone showed partial inhibition of tumor growth, two out of the five mice receiving the combination therapy showed complete tumor eradication (*Figure 6D*). In addition, none of the mice in this experiment seemed to display any adverse effects with clinical symptoms. These results suggested that the

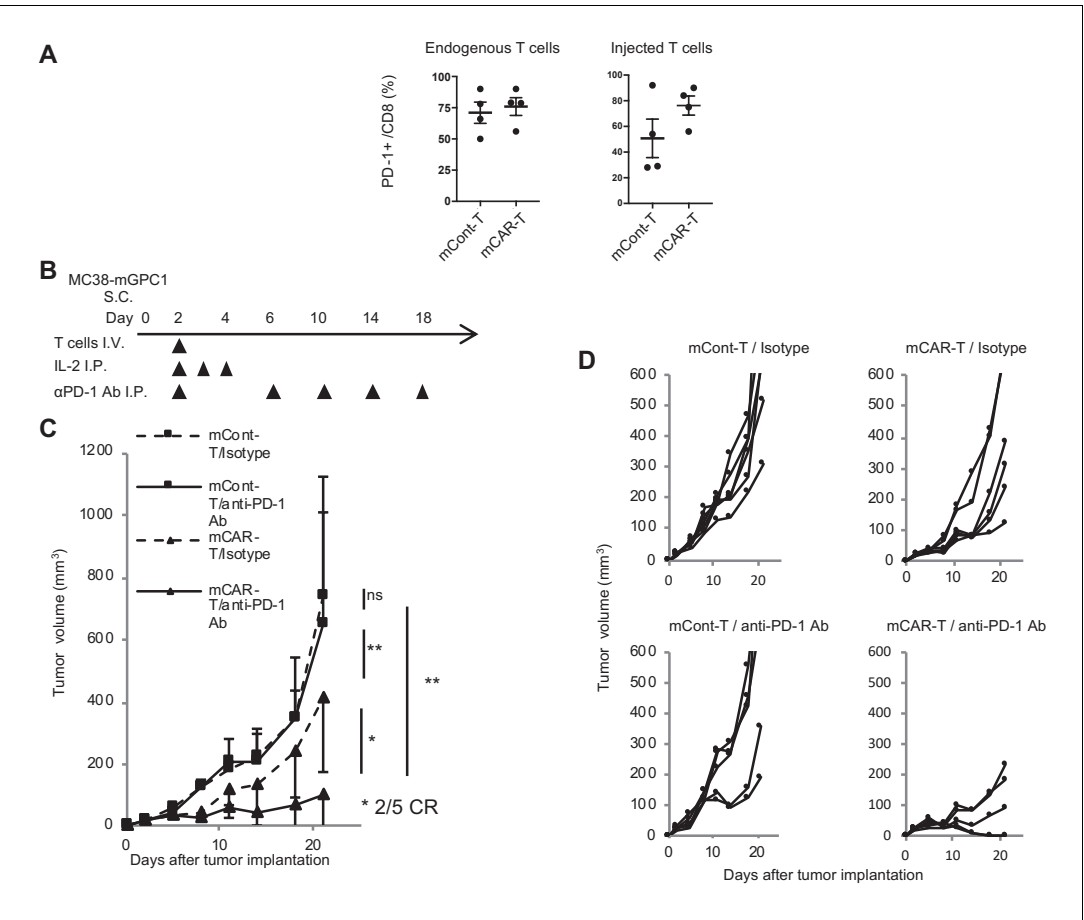

**Figure 6.** GPC1-specific mCAR-T cells synergized with anti-PD-1 Ab. (**A**) Percentages of PD-1-expressing cells among endogenous CD8[+] T cells (left panel) and injected CD8[+] T cells (right panel) harvested from the tumor tissues of MC38-mGPC1 mouse model 12 days after mCAR-T cell or mCont-T cell administration. Dots indicate individual mice of each group. (**B**) The protocol of combination therapy is shown. Mice bearing MC38-mGPC1 were treated with $2 \times 10^6$ cells of mCAR-T cells or mCont-T cells on day 2 and anti-PD-1 Ab (200 mg/mouse) or isotype Ab were intraperitonealy injected on days 2, 6, 10, 14, and 18. (**C**) Mean tumor volume (mm$^3$ ± SD) of each group is shown. (**D**) Tumor-growth curves of the individual mice in each group are shown. Results are representative of two or three experiments. Error bars indicate SD.

addition of anti-PD-1 Ab synergistically enhanced antitumor effects of the GPC1-specific CAR-T cells.

## Discussion

In this study, we developed GPC1-specific CAR-T cells for both human and mouse by using the scFV of anti-GPC1 mAb (clone: 1–12) that recognized both human and mouse GPC1. The hCAR-T cells specifically recognized and controlled tumor growth in xenograft mouse models, and the mCAR-T cells completely eradicated established solid tumor in vivo without any histological or symptomatic adverse effects in syngeneic mouse models. In addition, immune response against non-GPC1 endogenous tumor antigen was enhanced in syngeneic mouse models, suggesting the role of antigen spread induced by the CAR-T cell therapy in promoting further antitumor activity. Moreover, we were able to demonstrate synergistic antitumor effects by combining the CAR-T cell therapy with anti-PD-1 Ab therapy. Immunogenicity of the chicken-based CAR-T cells may cause rejection of the CAR-T cells and poor responses in patients. However, non-humanized CD19-specific CAR-T cells have demonstrated significant antitumor effects for a variety of B cell malignancy possibly due to

their relatively immunosuppressed condition via malignancy and chemotherapy/preconditioning. And the chicken-based GPC1-specific CAR-T cells showed strong antitumor effects in this study. Therefore, this form of CAR-T cells could be used for clinical trials, although we are also considering further humanization for clinical trials. Our preclinical study suggests that GPC1-specific CAR-T cells using the scFV of anti-GPC1 mAb (clone: 1–12) may provide not only safe but also highly effective treatment for patients with solid tumor.

Others and we have reported high expression of GPC1 in various solid tumors. On the other hand, we detected low expression of hGPC1 mRNA in all tested normal tissues and we also previously reported that some normal tissues showed slight staining by the commercial polyclonal anti-GPC1 Ab in IHC. In this study, we evaluated hGPC1 protein expressions by IHC using our anti-GPC1 mAb (clone: 1–12) and found that GPC1 expression was under the detection sensitivity in human normal tissues, whereas human esophageal SCC tissues showed strong and broad expression. The difference in IHC results of commercial polyclonal Ab and our anti-GPC1 mAb might be related to the difference of epitopes recognized by the antibodies. Although it is still uncertain whether GPC1 protein expression is completely negative in normal tissues of human, our anti-GPC1 mAb (clone: 1–12) specifically recognizes GPC1 protein expressed on tumor cells and is suitable for the generation of CAR-T cells.

A major mechanism of on-target off-tumor adverse effects is thought to be the recognition of target antigens expressed in normal cells by CAR-T cells (*Tran et al., 2013*; *Lamers et al., 2006*). However, some CAR-T cell therapies in clinical trials are targeting antigens that are expressed in both tumor and normal tissues (e.g. mesothelin, GD2, and IL13Rα2) and are demonstrating favorable responses without obvious adverse effects (*Beatty et al., 2014*; *Brown et al., 2016*; *Rossig et al., 2002*). For example, although mesothelin is broadly expressed on mesothelial cells at protein levels, patients treated with mesothelin-specific CAR-T cells show objective antitumor responses (*Beatty et al., 2014*; *Morello et al., 2016*). These results suggest the presence of unknown mechanisms of CAR-T cells in recognizing antigens in a tumor-specific manner. In our study, tumor-specific killing by our CAR-T cells may be explained by a tumor-specific recognition by the anti-GPC1 mAb (clone: 1–12). Another possibility may be that, even if GPC1 is expressed on some normal tissues and can be recognized by anti-GPC1 mAb (clone: 1–12), its expression might be too low to activate GPC1-specific CAR-T cells. Indeed, our GPC1-specific human or murine CAR-T cells did not respond to LK2-mock or MC38-mock tumor cells (*Figures 2C, D*, *4C and D*), although LK2-mock weakly expressed *hGPC1* mRNA (*hGPC1/GAPDH*; 0.00059) at similar level to the normal cervix tissues (*hGPC1/GAPDH*; mean 0.00209 ± 0.00185, range 0.00065–0.00473) and MC38-mock expressed weak *mGpc1* mRNA (*mGpc1/Gapdh*; 0.00532) at similar level to the evaluated various normal tissues (*mGpc1/Gapdh*; mean 0.02284 ± 0.01684, range 0.00341–0.06406) without detectable GPC1 protein (*Figures 1*, *2B* and *3*). In addition, some reports demonstrated that CAR-T cells have thresholds of antigen density required to induce responses against target antigens (*James et al., 2010*; *James et al., 2008*; *Walker et al., 2017*). Other possible explanation is the existence of anatomical barriers or deficiency of chemokines for the attraction of T cells into normal tissues (*Beatty et al., 2014*). Further studies are needed to clarify the mechanisms behind the lack of adverse effects in our models.

With exception of few studies using syngeneic immune-competent mouse, most previous preclinical studies have mainly used in vitro assays or xenogeneic immune-deficient mouse models for the evaluation of CAR-T cell function (*Carpenito et al., 2009*; *Gao et al., 2014*; *Jiang et al., 2016*; *Shiina et al., 2016*). Rosenberg et al. reported the use of syngeneic mouse models for the evaluation of CAR-T cells targeting VEGFR2 and FAP expressed in the stromal cells in tumor microenvironment rather than the tumor cells themselves (*Chinnasamy et al., 2010*; *Tran et al., 2013*). Although VEGFR2-specific CAR-T cells exhibited effective antitumor activity without any adverse effects, FAP-specific CAR-T cells resulted in lethal bone toxicities. These findings suggested the benefits of utilizing syngeneic mouse models for assessing the possible adverse effects of CAR-T cells. In this study, we first evaluated the efficacy and safety of CAR-T cells therapy targeting the tumor-expressing antigen using syngeneic mouse models. We generated GPC1-specific CAR-T cells using the scFv derived from anti-GPC1 mAb, which cross-reacted with both human and mouse GPC1. With the use of syngeneic mouse model, we demonstrated in vivo that our GPC1-specific CAR-T cells exhibited antitumor activity without any obvious adverse effects. These results suggest that our GPC1-specific CAR-T cells may become an effective treatment for patients with GPC1-positive tumor.

Syngeneic mouse model also allows us to evaluate the interaction of CAR-T cells and endogenous host immunity. We demonstrated that CAR-T cells enhanced the CTL induction and generated immunological memory against non-GPC1 endogenous tumor antigens. These findings might explain stronger antitumor effects of mCAR-T cells in the immunocompetent syngeneic mouse models compared to the delayed antitumor effects of hCAR-T cells in the immunodeficient xenogeneic mouse model. We speculated that lysis of tumor cells and secretion of multiple cytokines by CAR-T cells created an optimal environment for the activation of dendritic cells and CTL. Such kinds of phenomena were actually reported in some clinical trials. For example, induction of antitumor antibodies against novel antigens was reported in clinical trials of mesothelin-specific CAR-T cells (*Beatty et al., 2014*). Development of adaptive immunity against a broad spectrum of tumor-specific antigens by antigen spread is an important secondary mechanism underlying the potency of immunotherapy, which could overcome immune escape of the CAR-T cell therapy targeting single antigen (*Galon and Bruni, 2019*; *Kato et al., 2017*). In order to promote antitumor effects of antigen spread, it would be important to optimize pre- and post-conditioning protocols for CAR-T cell therapy by, for instance, lymphodepeletion and cytokine administration. Syngeneic mouse models for CAR-T cell therapies would provide important insights into the interaction of CAR-T cells and endogenous immunity.

Through the syngeneic mouse models, we can also evaluate the efficacy of combination immunotherapy in physiological condition, which cannot be evaluated precisely in xenogeneic immunodeficient mouse models because of their immunodeficiency and altered CAR-T cell phenotypes, which are activated by xenogeneic stimuli. For example, delayed antitumor effects observed in the NOG mouse model (*Figure 2E*) compared to syngeneic mouse models (*Figures 4E, F and G*) might be explained by xenogeneic GVHD reaction that might promote late activation and expansion of hCAR-T cells. In the syngeneic mouse models, strong PD-1 expression on the administered CAR-T cells and endogenous TIL, which suggested occurrence of T cell activation and exhaustion (*Iwai et al., 2017*), prompted us to evaluate synergistic effects of CAR-T cells and anti-PD-1 Ab therapies in our syngeneic mouse model. We found that the combination therapy led to synergistic antitumor effects without any obvious adverse effects. This synergistic effect may be explained by enhanced effector function of administered CAR-T cells and endogenous antitumor CTL by anti-PD-1 Ab. In addition, as PD-1 is reported to be involved in the suppression of CD28/B7 signal, it might be also possible that the anti-PD-1 Ab recovered the suppressed CD28 signal from dendritic cells during the T cell priming phase and consequently led to enhanced CTL induction (*Hui et al., 2017*; *Kamphorst et al., 2017*). The expression of PD-1 on CAR-T cells pre- and post-infusion was reported in human clinical trials and high frequency of PD-1 expression was associated with treatment failure (*Fraietta et al., 2018*; *O'Rourke et al., 2017*). In the clinical trials of EGFRvIII-specific CAR-T cells for glioma patients, upregulation of PD-L1 expression on tumor cells after infusion of CAR-T cells was observed in non-responder patients. These reports suggest that PD-1 is a potential target to improve the efficacy of CAR-T cell therapies and syngeneic mouse models are effective tools to evaluate the effectiveness of combination therapy with CAR-T cells.

Collectively, we have generated CAR-T cells targeting GPC1 of both human and mouse, and shown their efficacy in xenograft mouse models. By establishing the syngeneic mouse models, we were able to evaluate not only the efficacy but also the safety of GPC1-specific CAR-T cells and their enhanced antitumor activity in combination with anti-PD-1 Ab. Taken together, we have demonstrated the strong antitumor effects and safety of GPC1-specific CAR-T cells against GPC1-expressing solid tumors using both syngeneic and xenogeneic mouse models.

## Materials and methods

**Key resources table**

| Reagent type (species) or resource | Designation | Source or reference | Identifiers | Additional information |
|---|---|---|---|---|
| Strain, strain background (*M. musculus*) | NOG (NOD/Shi-*Prkdc*^scid^ *Il2rγ*^tm1Sug^/Jic) mouse | Central Institute for Experimental Animals | | Female and Male, 6–10 week-old |

*Continued on next page*

*Continued*

| Reagent type (species) or resource | Designation | Source or reference | Identifiers | Additional information |
|---|---|---|---|---|
| Strain, strain background (*M. musculus*) | C57BL/6 mouse | CLEA Japan, Inc | | Female, 6–8 week-old |
| Cell line (*Homo-sapiens*) | G3Thi | Takara Bio Inc | TKR-6163 | Human kidney cell line derived packaging cell line |
| Cell line (*Homo-sapiens*) | TE14 | RIKEN Bio Resource Center | | Human esophageal squamous cancer cell lines |
| Cell line (*Homo-sapiens*) | LK2-hGPC1 | Previous paper (*Harada et al., 2017*), see Materials and methods | | Human lung squamous cancer cell lines expressing hGPC1 |
| Cell line (*Homo-sapiens*) | LK2-mock | Previous paper (*Harada et al., 2017*), see Materials and methods | | Human lung squamous cancer cell lines expressing empty vector |
| Cell line (*M. musculus*) | PG13 | Takara Bio Inc | | Murine leukemia cell line |
| Cell line (*M. musculus*) | EL4 | National Cancer Institute, National Institutes of Health (MD, USA) | | Mouse T cell lymphoma cell line |
| Cell line (*M. musculus*) | MCA205 | National Cancer Institute, National Institutes of Health (MD, USA) | | Mouse sarcoma cell line |
| Cell line (*M. musculus*) | MCA205-mGPC1 | This paper, see Materials and methods | | Mouse sarcoma cell line expressing mGPC1 |
| Cell line (*M. musculus*) | MC38-mGPC1 | This paper, see Materials and methods | | Mouse colon adenocarcinoma cell line expressing mGPC1 |
| Transfected construct (Gallus gallus domesticus, *Homo-sapiens*) | hCAR-LH | This paper, see Materials and methods and *Figure 2—figure supplement 1* | | |
| Transfected construct (Gallus gallus domesticus, *Homo-sapiens s*) | hCAR-HL | This paper, see Materials and methods and *Figure 2—figure supplement 1* | | |
| Transfected construct (Gallus gallus domesticus, *M. musculus*) | mCAR | This paper, see Materials and methods and *Figure 4—figure supplement 1* | | |
| Antibody | OKT-3 (anti-human CD3 mAb) | Thermo Fischer Scientific | Cat # 16-0037-81, RRID:AB_2619696 | T cell activation, 50 ng/ml |
| Antibody | Anti-mouse PD-1 Ab (clone: J43) | Bio X Cell | BE-0033–2, RRID:AB_1107747 | In vivo injection, 200 mg/mouse |
| Antibody | Anti-human IFNγ Ab (M700A) | Thermo Fisher Scientific | Cat # M700A, RRID:AB_223578 | For ELISA |
| Antibody | Anti-human IFNγ Ab (M700B) | Thermo Fisher Scientific | Cat # M700B | For ELISA |
| Antibody | anti-GPC1 Ab (clone: 1–12) | previous paper (*Harada et al., 2017*) | | IHC-F(0.5 µg/ml) |
| Antibody | anti-GFP mAb (clone: 1E4) | Medical and Biological Laboratories | Code # M-048–3, RRID:AB_591823 | IHC-P(0.5 µg/ml) |

*Continued on next page*

*Continued*

| Reagent type (species) or resource | Designation | Source or reference | Identifiers | Additional information |
|---|---|---|---|---|
| Antibody | APC-donkey-anti-IgY Ab | Jackson immuno Research Inc | Code # 703-136-155, RRID:AB_2340360 | FACS (1:50) |
| Antibody | PE-donkey-anti-IgY Ab | Jackson immuno Research Inc | Code # 703-116-155, RRID:AB_2340358 | FACS (1:50) |
| Antibody | V500- anti-mCD45 mAb | BD Biosciences | Cat # 561487, RRID:AB_10697046 | FACS (1:50) |
| Antibody | BV421-anti-mCD3 mAb | Biolegend | Cat # 562600, RRID:AB_11153670 | FACS (1:50) |
| Antibody | Alexa Fluor 700-anti-mCD8 mAb | BD Biosciences | Cat # 557959, RRID:AB_396959 | FACS (1:50) |
| Sequence-based reagent | human GPC-1 (Hs00892476_m1) | Applied Biosystems | Cat # 4331182 | |
| Sequence-based reagent | mouse GPC-1 (Mm00497305_m1) | Applied Biosystems | Cat # 4331182 | |
| Peptide, recombinant protein | Human GPC1 | Biolegend | Cat # 757206 | |
| Peptide, recombinant protein | Mouse GPC1 | Biolegend | Cat # 757306 | |
| Peptide, recombinant protein | MuLV gp70 p15E | Medical and Biological Laboratories | Code # TS-M507-P | KSPWFTTL |
| Peptide, recombinant protein | H-2Kb-restricted b-galactosidase | Medical and Biological Laboratories | Code # TS-M501-P | DAPIYTNV |
| Commercial assay or kit | Series S Sensor Chip CM5 | GE Healthcare Life Sciences | Cat # BR-1005–30 | |
| Commercial assay or kit | Mouse Antibody Capture Kit | GE Healthcare Life Sciences | Cat # BR1008-38 | |
| Commercial assay or kit | Mouse IFNγ ELISA set | BD Biosciences | Cat # 555138 | |
| Software, algorithm | Kaluza 1.2 | BECKMAN COULTER | | |
| Software, algorithm | GraphPad Prism 7.0 | GraphPad software | | |
| Other | FDA Standard Frozen Tissue Array-Human Adult Normal | BioChain Institute Inc | Cat # T6234701-1 | |
| Other | human normal cervix tissues and cervical carcinoma tissues | This paper, see Materials and methods | | |
| Other | Cr$^{51}$ | Japan Radioisotope Association | | |

## Tissue samples

The frozen tissue arrays of normal human tissues (FDA Standard Frozen Tissue Array – Human Adult Normal) were purchased from BioChain Institute Inc, CA. The human normal cervix tissues and cervical carcinoma tissues were surgically resected from the patients at Keio University hospital. The frozen tissue samples of normal mice (C57BL/6) were made by a general protocol. For the evaluation of adverse effects, the normal tissues of T cells transferred mice were prepared by perfusion fixation using 4% paraformaldehyde. These tissues were then formalin-fixed and paraffin-embedded.

## Quantitative reverse transcription PCR analysis

Total RNA was isolated from tissues using RNeasy columns (Qiagen, Hilden, Germany) and reversed transcribed using ReverTra Ace qPCR RT Master Mix with gDNA Remover (Toyobo Co., Osaka, Japan) according to the manufacturer's instructions. Quantitative reverse transcription PCR (RT-qPCR) was performed using THUNDERBIRD Probe qPCR Mix (Toyobo Co.) and TaqMan probes (Applied Biosystems). GAPDH was used as a housekeeping gene for quantitative real-time PCR normalization (cat # 4310884E for human and # 4352339E for mouse, Applied Biosystems, CA). TaqMan RT-PCR (Applied Biosystems) probes were human GPC-1 (Hs00892476_m1) and mouse GPC-1 (Mm00497305_m1).

## Immunohistochemistry

Immunohistochemistry (IHC) was done on frozen sections (for GPC1) or formalin-fixed paraffin-embedded sections (for GFP) of human and mouse systemic tissues using protocols previously described (*Nakamura et al., 2018*). Anti-GPC1 chicken/mouse chimeric mAb (clone: 1–12, 0.5 µg/ml) were generated by immunization of consensus region of human and mouse GPC1 as previously described (*Harada et al., 2017*). Anti-GFP mAb (clone: 1E4, 0.5 µg/ml) was purchased from Medical and Biological Laboratories (Aichi, Japan). Histofine simple stain MAX-PO (Nichirei Biosciences Inc, Tokyo, Japan) was used for detection of primary antibodies in human tissues and Histofine mouse stain kit (Nichirei Biosciences Inc) was used for detection of primary antibodies in mouse tissues according to the manufacturer's instructions.

## Cell lines and media

Human esophageal squamous cancer cell lines (TE14) were obtained from the RIKEN BioResource Center (Ibaraki, Japan) and transferred to the Keio University School of Medicine in 2015. Human lung squamous cancer cell lines were obtained from the Japanese Collection of Research Bioresources (Osaka, Japan), generated the enforced expression of human GPC1 (LK2-hGPC1) or empty vector (LK2-mock) cell lines as previously described (*Harada et al., 2017*), and transferred to the Keio University School of Medicine in 2015. Mouse colon adenocarcinoma cell line (MC38), mouse sarcoma cell line (MCA-205), and mouse T cell lymphoma cell line (EL4) were obtained from the Surgery Branch of the National Cancer Institute, National Institutes of Health, MD in 1998, where MC38 and MCA-205 were developed. MC38 and MCA205 cells were transduced with lentivirus vectors encoding mouse GPC1 (mGPC1) cDNA as previously described (*Yaguchi et al., 2012*). MC38 and MCA205 stably expressing mGPC1 are designated as MC38-mGPC1 and MCA205-mGPC1, respectively. The identity of each human cell line was confirmed by DNA fingerprinting *via* short tandem repeat (STR) profiling as previously described (*Harada et al., 2017*). Each cell line was thawed from lab frozen stock which were generated from early passages and utilized for each experiment within 4 weeks culture. All the cell lines utilized were Mycoplasma free determined by qPCR analysis.

Human T cells were cultured in AIM-V (Thermo fisher Scientific, MA) containing 10% heat-inactivated human AB serum (Gemini Bio-Products, CA) and 300 IU/ml recombinant human interleukin-2 (rhIL-2) (Novartis, NJ). Mouse T cells were cultured in RPMI1640 (Thermo Fischer Scientific) containing 10% heat-inactivated FBS, 100 U/ml penicillin, 100 µg/ml streptomycin, 0.05 mM 2-mercaptoethanol, 0.1 mM MEM nonessential amino acids, 1 mM sodium pyruvate, and 10 mM l-HEPES (all from Thermo Fischer Scientific), and 50 IU/ml of rhIL-2 (Novartis).

## Retrovirus vector designs

The sequences encoding the anti-GPC1 single-chain variable fragment (scFv) of VL-VH (LH form) or VH-VL (HL form) are based on the sequence of anti-GPC1 mAb (clone: 1–12) that recognize human and mouse GPC1 (*Harada et al., 2017*). As shown in *Figure 2A* and *Figure 2—figure supplement 1*, the human CAR (hCAR) comprising the scFv GPC1 linked to the human CD8a leader sequence (nucleotides 1–63, GenBank NM 001768.6), human CD28 extracelluaar/transmembrane/intracellular domains (nucleotides 562–882, GenBank NM 001768.6), and human CD3ζ intracellular domain (nucleotides 299–637, GenBank NM_000734.3). For generation of murine CAR (mCAR), as shown in *Figure 4A* and *Figure 4—figure supplement 1*, human CD8, CD28, and CD3ζ sequences were converted to mouse CD8 (nucleotides 1–81, GenBank NM_001081110.2), mouse CD28 (nucleotides 429–740, GenBank NM_007642.4), and mouse CD3ζ (nucleotides 302–643, GenBank) sequences,

respectively. These hCAR and mCAR genes were cloned in-frame into the pMS3-F retroviral vector (Takara Bio Inc, Shiga, Japan).

## Generation of human and murine CAR-T cells

Transient retroviral supernatants were generated by co-transfecting G3Thi cells (Takara Bio Inc) with the human or murine CAR plasmid, the ECO envelope plasmid, and gag-pol plasmid (all form Takara Bio Inc), using Hily Max (Dojindo Laboratories, Kumamoto, Japan). After 12 hr, supernatants were replaced by fresh medium and retroviral supernatants were collected at 24 hr after replacement of medium. The generated murine GPC-1 CAR-T (ECO Env.) retroviral vector was used for the generation of murine CAR-T cells. The human GPC-1 CAR-T (ECO Env.) retrovirus vector was introduced into PG13 cells (Takara Bio Inc) for generating the GaLV Env. retroviral vector for the generation of human hCAR-T cells. These retroviral supernatants were centrifuged onto retronectin (Takara Bio Inc) -coated plates at 2000 g for 2 hr at 32˚C as previously descried (*Inozume et al., 2016*).

For the activation of human T cells, PBMCs from healthy donors were stimulated with soluble 50 ng/ml OKT-3 (Thermo Fischer Scientific) for 2 days before transduction. The stimulated cells were then transduced by spin-down onto the retrovirus plates for 10 min at 1000 g. For murine T cells, splenocytes were collected from transgenic mice that ubiquitously express EGFP under control of the CAG promoter (*Kawamoto et al., 2000*) and activated on day 0 with 2.5 mg/mL concanavalin A (Sigma-Aldrich, MO) supplemented with 1 ng/mL rmIL-7 (Peprotech, NJ) for 1 day before transduction. The stimulated cells were then transduced by spin-down onto the retrovirus plates for 10 min at 1000 g. GPC1-specific human CAR gene (LH or HL) and murine CAR (HL) gene transduced into human and murine stimulated T cells, respectively.

## Flowcytometeric analysis and cell isolation

Tumor cell lines were stained by anti-GPC1 Ab (clone: 1–12) as previously described (*Harada et al., 2017*). The surface expression of CAR on the transduced T cells was evaluated by APC-donkey-anti-IgY Ab (Dilution 1:50, Jackson immunoResearch Inc, PA, USA). For IFNγ-releasing assay and $Cr^{51}$ releasing assay of hCAR-T cells, hCAR expressing T cells were positively isolated using PE-donkey-anti-IgY Ab (Jackson immunoResearch Inc) and anti-PE microbeads (Miltenyi Biotec, Bergisch Gladbach, Germany) according to the manufacturer's instructions. Mice cells in the tissues and the peripheral blood were stained with anti-CD45 (Dilution 1:50, Fluorophore V500, Clone 30-F11, BD Biosciences, NJ), anti-CD3 (Dilution 1:50, Fluorophore BV-421, Clone 145–2 C11, Biolegend, CA), and anti-CD8 (Dilution 1:50, Fluorophore Alexa Fluor 700, Clone 53–6.7, BD Biosciences) Abs for 1 hr at 4˚C and then washed three times with FACS buffer (2% FCS PBS).

## Cytokine secretion assay

Cultured tumor cell lines were used as stimulator cells ($5 \times 10^4$ cells/well). T cells ($1 \times 10^5$ cells/well) were co-cultured with the stimulators in a 96-well plate and supernatants were harvested after 24 hr. Human (M700A and M701B; Endogen) and murine (BD Biosciences) IFNγ were measured by ELISA.

## Chromium release assay

Target cells were loaded with $Cr^{51}$ (Japan Radioisotope Association, Tokyo, Japan) and co-cultured with differing amounts of CAR-T cells. After a 4 hr incubation at 37˚C, the release of free $Cr^{51}$ was measured by TopCount NXT (PerkinElmer Inc, MA) as previously described (*Yaguchi et al., 2012*). The percent-specific lysis was calculated using the formula: % specific lysis = 100 x (experimental cpm release – spontaneous cpm release)/(total cpm release – spontaneous cpm release). All data are represented as a mean of triplicate wells (± SD).

## Surface Plasmon resonance (SPR) analysis

The sequences of anti GPC-1 scFv were cloned into pCAG-Neo mIgG2a-Fc plasmid (FUJIFILM Wako, Osaka, Japan) for the generation of anti GPC1_scFv-mIgG2a_Fc fusion protein. The plasmids were transfected into 293 T cells and the fusion proteins were purified from their culture supernatants using Antibody TCS Purification Kit (Abcam, Cambridge, UK). The binding affinity of LH form and HL form of anti GPC1 scFv-mIgG2a Fc fusion protein to recombinant hGPC1 (Biolegend) and recombinant mGPC1 (Biolegend) was assessed by SPR using Biacore T200 (GE Healthcare Life

Sciences, Tokyo, Japan). Measurement was performed at 25°C in HBS-EP+ buffer. Series S Sensor Chip CM5 (GE Healthcare Life Sciences) was immobilized with polyclonal rabbit anti-mouse IgG using Mouse Antibody Capture Kit (GE Healthcare Life Sciences) and Amine Coupling Kit (GE Healthcare Life Sciences) according to the manufacturer's protocol. scFv-Fc fusion protein (1 µg/ml) was injected at 10 µl/min for 1 min and captured by the immobilized anti-mouse IgG. Next, association and dissociation kinetics was monitored at a flow rate of 30 µl/min. Two-fold dilution series of recombinant hGPC1 (6.25–400 nM) or mGPC1 (25–400 nM) was injected for 2 min. Dissociation was monitored for 7.5 min. The sensor chip was regenerated by 10 mM glycine–HCl (pH 1.7) for 0.5 min. Binding to the sensor chip is given as resonance units (RU). Data were analyzed by Biacore T200 Evaluation Software (v2.0) (GE Healthcare Life Sciences).

## Mouse model analysis

Mice were bred at the animal facilities of Keio University according to guidelines for animal experimentation.

For xenogeneic mouse model, $3 \times 10^6$ TE14 cells were subcutaneously inoculated into the flank of 6–10 week-old NOG (NOD/Shi-$Prkdc^{scid}$ $Il2r\gamma^{tm1Sug}$/Jic) mice. Cultured hCAR-T cells or human control T (hCont-T) cells ($2 \times 10^7$ cells per mouse) were intravenously administered on day 9.

For syngeneic mouse models, $5 \times 10^5$ MC38-mGPC1 or MCA205-mGPC1 cells were subcutaneously inoculated into the flank of 6–8 weeks old C57BL/6 mice and conditioned for adoptive cell therapies as previously reported (Chinnasamy et al., 2010; Tran et al., 2013). On day 2–3, the mice were conditioned with 5 Gy total body irradiation (TBI) immediately before the T-cell transfer and the cultured mCAR-T cells or murine control T (mCont-T) cells ($2 \times 10^6$ cells per mouse) were intravenously administered. Subsequently, the mice were given intraperitoneal injections of 50,000 IU/mouse rhIL-2 twice daily up to six doses. For combination therapy model, either 200 mg/mouse anti-PD-1 (clone J43, Bio X Cell, NH) or isotype antibody (clone PIP, Bio X Cell) was intraperitonealy injected on days 2, 6, 10, 14, and 18. For the re-challenge model, 120 days after the mCAR-T cell administration, $5 \times 10^5$ parental MCA205 cells were subcutaneously inoculated into opposite site of the flank in naive mice with no history of bearing tumors or the mice which had rejected MCA-205-mGPC1 by the mCAR-T cell therapy. Tumor volumes were calculated according to the following formula: [(length) × (width)2]/2.

## Evaluation of Tumor-Specific CD8+ T cell responses

To evaluate CD8+ tumor-infiltrating lymphocytes (TIL) immune responses specific for endogenous tumor antigens, CD8+ T cells were magnetically sorted (Miltenyi Biotec) from the tumor, cocultured with syngeneic 32Gy irradiated splenocytes in RPMI1640 with 10% FBS, and restimulated with 1 µg/mL gp70 peptide, the H-2K$^b$-restricted T cell epitope peptide (MuLV gp70 p15E; aa 604-611 (KSPWFTTL)). After 2 days restimulation, T cells were collected using Lympholyte-M Cell Separation Media (Cedarlane, Ontario, Canada), cocultured with EL4 pulsed with 1 µg/ml gp70 peptide or control peptide (H-2Kb-restricted b-galactosidase, Medical and Biological Laboratories) for 24 hr, and evaluated IFNγ secretion by ELISA (BD Biosciences) as described previously (Kudo-Saito et al., 2009).

## Statistical analysis

All results are shown as mean ± SD. Data were subjected to statistical analysis (unpaired t test and Bonferroni/Dunn's test) to determine significant differences between the means of experimental and control groups. GraphPad Prism 7.0 was used for the statistical calculations. $p<0.05$ (*) and $p<0.01$ (**) were considered statistically significant.

## Acknowledgements

This work was supported by Grants-in-aid for Scientific Research from the Ministry of Education, Culture, Sports, Science and Technology (MEXT) of Japan (26221005); the Project for Development of Innovative Research on Cancer Therapeutics (P-DIRECT) (14069014) and the Project for Cancer Research And Therapeutic Evolution (P-CREATE) from Japan Agency for Medical Research and Development (AMED); a grant from Tokyo Biochemical Research Foundation; a Keio Gijuku Academic Development Funds.

We would like to thank Miyuki Saito for technical assistance and JeongHoon Park, Misako Horikawa, and Ryoko Suzuki for preparation of the manuscript.

## Additional information

### Funding

| Funder | Grant reference number | Author |
|---|---|---|
| Ministry of Education, Culture, Sports, Science, and Technology | Grants-in-aid for Scientific Research 262221005 | Yutaka Kawakami |
| Japan Agency for Medical Research and Development | Project for Development of Innovative Research on Cancer Therapeutic 14069014 | Tomonori Yaguchi |
| Japan Agency for Medical Research and Development | Project for Cancer Research And Therapeutic Evolution | Yutaka Kawakami |
| Tokyo Biomedical Research Foundation | Research Grant | Yutaka Kawakami |
| Keio University | Keio Gijuku Academic Development founds | Tomonori Yaguchi |

The funders had no role in study design, data collection and interpretation, or the decision to submit the work for publication.

### Author contributions

Daiki Kato, Conceptualization, Data curation, Formal analysis, Validation, Investigation, Visualization, Methodology, Project administration; Tomonori Yaguchi, Conceptualization, Data curation, Formal analysis, Supervision, Funding acquisition, Validation, Investigation, Visualization, Methodology, Project administration; Takashi Iwata, Formal analysis, Validation, Visualization, Methodology; Yuki Katoh, Kinya Tsubota, Yoshiaki Takise, Masaki Tamiya, Data curation, Formal analysis, Validation, Visualization, Methodology; Kenji Morii, Data curation, Formal analysis, Validation, Visualization, Methodology, Project administration; Haruhiko Kamada, Hiroki Akiba, Kouhei Tsumoto, Resources, Formal analysis, Methodology; Satoshi Serada, Tetsuji Naka, Ryohei Nishimura, Takayuki Nakagawa, Resources, Methodology, Project administration; Yutaka Kawakami, Conceptualization, Resources, Software, Supervision, Funding acquisition, Validation, Investigation, Visualization, Methodology, Project administration, Writing - review and editing

### Author ORCIDs

Daiki Kato (iD) https://orcid.org/0000-0002-0964-4158
Tomonori Yaguchi (iD) https://orcid.org/0000-0002-2904-9030
Kinya Tsubota (iD) http://orcid.org/0000-0002-7680-3745
Yutaka Kawakami (iD) https://orcid.org/0000-0003-4836-2855

### Ethics

Human subjects: All human sample donors were provided explanatory materials and a verbal explanation of the procedure, detailing both the procedure and the purposes of the experiment, as well as their rights, prior to collection and use. All experimental procedures were reviewed and approved by the Keio University School of Medicine Ethics committee (Approval Number: 20110159 and 20130122).
Animal experimentation: All experimental procedures were reviewed and approved by the Keio University Institutional Animal Care and Use Committee (Approval Number: 09037).

### Decision letter and Author response

Decision letter https://doi.org/10.7554/eLife.49392.sa1
Author response https://doi.org/10.7554/eLife.49392.sa2

## Additional files

### Supplementary files
• Transparent reporting form

### Data availability
All data generated or analysed during this study are included in the manuscript and figures.

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
