## [Decision Letter]

**Acceptance summary:**

Your finding on combination therapy of CAR-T and anti-PD-1 will trigger a new dimension in the field of cancer immunotherapy.

**Decision letter after peer review:**

Thank you for submitting your article "GPC1 specific CAR-T cells eradicate established solid tumor without adverse effects and synergize with anti-PD-1 Ab" for consideration by *eLife*. Your article has been reviewed by two peer reviewers, and the evaluation has been overseen by a guest Reviewing Editor and Jeffrey Settleman as the Senior Editor. The reviewers have opted to remain anonymous.

Although the CAR-T cell treatment has been very effective for leukemia, it has been difficult for the treatment of solid tumors. The authors report that CAR-T cell using anti-GPC1 monoclonal antibody, which reacts with both human and the mouse GPC1, can serve a good reagent for the solid tumor treatment. They showed significant anti-tumor effects in the xenogeneic and syngeneic mouse models. They have also shown this CAR-T cells have minimal side effects. However, this study contains significant shortcoming and the experimental problems to convincingly prove that CAR-T cell is effective for solid tumors.

The reviewers have discussed the reviews with one another and the Reviewing Editor has drafted this decision to help you prepare a revised submission. The revisions proposed by the reviewers (below) should all be considered essential to support further consideration of the revised manuscript. We realize that the requested revisions will take several months to carry out and suggest that you take the time needed to revise the paper appropriately.

Reviewer #1:

The authors report that GPC1, which is expressed at high levels in several types of cancers but at low levels in normal tissues, is a good target for CAR T cell therapy. They established CAR T cells derived from the anti-GPC1 mAb (1-12), which reacts with both human and mouse GPC1. The GPC1-specific human and murine CAR-T cells showed significant anti-tumor effect in xenogeneic and syngeneic mouse model, respectively, without any side effect. This manuscript presents a novel candidate target for CAR T cells against solid tumors. In addition, they succeeded in showing the safety of the CAR T cells by establishing CAR T cells that react with both human and murine GPC1. Thus, I recommend Accept with Major Revisions.

1) In Figure 1A and Figure 3A, GPC1 expression levels in cancer cells should be also shown. It will be helpful for readers to understand the difference of GPC1 expression levels between normal cells and cancers.

2) GPC1 mRNA is also expressed in normal tissues, although at lower levels than tumor cells, according to the results (Figure 1A or Figure 3A). As discussed in the manuscript (Discussion, third paragraph), the GPC1 antigen densities in normal tissues may be under the threshold level for the CAR recognition. This hypothesis should be tested by experiments, and discuss why normal cells are not damaged by the CAR T cells.

3) In xenograft model (Figure 2E), the in vivo effect of the CAR T cells was statistically significant, but not strong. In addition, the difference in tumor sizes between the CAR group and the control group became significant at relatively late time points. In contrast, in the syngeneic model using murine CAR T cells, especially in the MCA-mGPC1-transplanted mice (Figure 4F), the effect of CAR T cells was striking. This discrepancy suggests that endogenous T cells (or other immune cells) have substantial roles in the elimination of tumor cells in the syngeneic model. The authors may be able to examine it by using mice lacking T cells such as Rag2-KO for syngeneic models. At least, the authors are recommended to discuss more about this point.

4) Expression level of GPC1 in LK2-hGPC1 shown in Figure 2B and that in Figure 4B looks different. This information is important to know how high GPC1 expression in the mouse tumor cells in which GPC1 is enforcedly expressed.

Reviewer #2:

In this study, the authors demonstrated that GPC1 could be a potential target of CAR-T cell therapy in some types of solid tumors. It was also shown that the anti-tumor efficacy of the anti-GPC1 CAR-T cells was augmented by the combination with anti-PD-1 antibody. Based on cross-reactivity of the anti-GPC1 Ab, they also proposed the safety of anti-GPC1 CAR-T cell therapy. While CAR-T cell therapy against solid tumors are of highly interest as the next generation of cancer immunotherapy, this study contains significant defects in the experimental data and conclusion led by the authors. Inclusion of additional data and discussions according to below comments are necessary.

1) The authors concluded the safety of anti-GPC1 CAR-T cell therapy based on the cross-reactivity of Ab to mouse GPC1 and the similarity of GPC1 expression patterns between mouse and human normal tissues. However, this conclusion is hardly acceptable due to following reasons: 1) Even though their anti-GPC1 Ab cross-reacts with mouse GPC1, detail characteristics of the protein interaction are not necessarily identical. For example, affinity, avidity, Kd, and Ka of the Ab to human GPC1 and mouse GPC1 might not be the same. Moreover, the CAR construct utilized scFv instead of full form of Ab. Thus, detail characteristics as above must be assessed by using scFv. 2) As anti-GPC1 Ab used in this study was generated in chicken, immunogenicity of anti-GPC1 scFv will not be identical in human and mouse. Due to these reasons, observations in this study are not enough to conclude the safety of anti-GPC1 CAR-T cell therapy in human.

2) As for anti-tumor effects of anti-GPC1 CAR-T cells, experimental models are not appropriate to evaluate its potency. In syngeneic mouse models, CAR-T cells were injected 3 days after tumor inoculation. Day 3 tumor size is too small to consider as pre-established tumor models. Experiments to treat late-stage tumors with larger size are necessary. In addition, 50,000 IU IL-2 was injected up to 6 doses after CAR-T cell therapy, which is unusual in mouse CAR-T experiments and will not be used in clinical settings. Anti-tumor efficacy of anti-GPC1 CAR-T cells without IL-2 injections are needed.

3) Epitope spreading following anti-GPC1 CAR-T therapy was suggested based on IFN-g responses to gp70 in the treated mice (Figure 4I). However, it remains unclear whether the epitope spreading plays a crucial role in the anti-tumor effects of anti-GPC1 CAR-T therapy or is merely a by-stander phenomenon. In order to address this important question, efficacy of anti-GPC1 CAR-T cells should be examined in the mice with a depletion (or deficient) of endogenous T cell (or CD8 T cells). In addition, experiments to re-challenge GPC1-negative parental tumor into the mice which had rejected GPC1-positive tumor by CAR-T therapy are important.

4) In NOG mouse model, CAR-T cells were injected 9 days after tumor inoculation, but the suppression of tumor growth was visible only after day 30 (Figure 2E). Please explain any reason why there is such a gap between CAR-T injection and the detection of anti-tumor efficacy. Were any GVHD symptoms such as body weight loss observed in these NOG mice with CAR-T treatment?

5) MC38 is known as PD-1 Ab treatment-sensitive tumor. In this study, however, control T + anti-PD-1 Ab did not show any significant anti-tumor effects compared to control T + isotype Ab (Figure 6). Potential reasons for this observation should be explained.

---

## [Author Response]

Reviewer #1:The authors report that GPC1, which is expressed at high levels in several types of cancers but at low levels in normal tissues, is a good target for CAR T cell therapy. They established CAR T cells derived from the anti-GPC1 mAb (1-12), which reacts with both human and mouse GPC1. The GPC1-specific human and murine CAR-T cells showed significant anti-tumor effect in xenogeneic and syngeneic mouse model, respectively, without any side effect. This manuscript presents a novel candidate target for CAR T cells against solid tumors. In addition, they succeeded in showing the safety of the CAR T cells by establishing CAR T cells that react with both human and murine GPC1. Thus, I recommend accept with major revisions.1) In Figure 1A and Figure 3A, GPC1 expression levels in cancer cells should be also shown. It will be helpful for readers to understand the difference of GPC1 expression levels between normal cells and cancers.

To clarify the different expression of GPC1 between cancer cells and normal cells, we performed additional qPCR experiments using human cervical carcinoma (n=48) and normal cervix tissues (n=4) as well as various normal tissues as shown in the previous version. The tumor tissues expressed hGPC1 mRNA higher than normal cervix tissues and other various normal tissues. We added the new figures as Figure 1A and B and sentences in the Results, Materials and methods and figure legend sections as follows,

Results: “The human normal cervix tissues and cervical carcinoma tissues were surgically resected from the patients at Keio University hospital.”

Materials and methods: “To confirm tumor specific expression of hGPC1, we first analyzed expression of hGPC1 mRNA in cervical squamous cell carcinoma tissues, various adult human normal tissues, and fetal brain tissues by qPCR analysis. Most of the cervical squamous cell carcinoma tissues expressed higher hGPC1 mRNA than corresponding normal cervix tissues and various normal tissues (Figure 1A and B).”

Figure 1A legend: “A) The mRNA expression of hGPC1 was evaluated by qPCR in human normal cervix and cervical squamous carcinoma tissues; GAPDH was used as an internal control.”

We did not compare mGPC1 expression between normal and tumor tissues, because we could not find murine tumor cell lines endogenously expressing mouse GPC1.

2) GPC1 mRNA is also expressed in normal tissues, although at lower levels than tumor cells, according to the results (Figure 1A or Figure 3A). As discussed in the manuscript (Discussion, third paragraph), the GPC1 antigen densities in normal tissues may be under the threshold level for the CAR recognition. This hypothesis should be tested by experiments, and discuss why normal cells are not damaged by the CAR T cells.

The human or murine CAR-T cells did not produce hIFNγ or kill cancer cells when co-cultured with LK2-mock or MC38-mock which shows weak GPC1 mRNA expression similar to various normal cells, but not detectable GPC1 protein by flow cytometric analysis. These observations indicate that GPC1-specific human and murine CAR-T cells do not respond to normal cells. We added the sentence in the Discussion sections as follows:

“Indeed, our GPC1-specific human or murine CAR-T cells did not respond to LK2-mock or MC38-mock cancer cells which show a weak GPC1 mRNA expression without detectable GPC1 protein (Figure 2C and D, Figure 4C and D), similar to various normal cells.”

3) In xenograft model (Figure 2E), the in vivo effect of the CAR T cells was statistically significant, but not strong. In addition, the difference in tumor sizes between the CAR group and the control group became significant at relatively late time points. In contrast, in the syngeneic model using murine CAR T cells, especially in the MCA-mGPC1-transplanted mice (Figure 4F), the effect of CAR T cells was striking. This discrepancy suggests that endogenous T cells (or other immune cells) have substantial roles in the elimination of tumor cells in the syngeneic model. The authors may be able to examine it by using mice lacking T cells such as Rag2-KO for syngeneic models. At least, the authors are recommended to discuss more about this point.

To evaluate the presence of endogenous T cell responses to endogenous tumor antigens in vivo, we performed re-challenge experiments; re-challenge of GPC1-negative parental MCA205 tumor cells in the mice which had complete eradication of MCA205-mGPC1 by the GPC1-specific mCAR-T cell therapy. The cured mice showed a significant growth inhibition of the re-challenged tumors, indicating the presence of strong endogenous tumor specific T cells which may be involved in the strong antitumor effects of CAR-T therapy in syngeneic murine tumor models. We added a new figure in Figure 4K and sentences in the Results, Materials and methods and figure legend sections as follows:

Results: “Moreover, to evaluate the presence and importance of endogenous T cell responses in vivo, mGPC1-negative parental MCA205 tumor cells were re-challenged into the mice which had complete tumor eradication by mCAR-T cell therapy in the MCA205-mGPC1 model (Figure 4F). These mice showed a significant growth inhibition of the inoculated mGPC1-negative tumors, indicating the induction and involvement of T cells specific for endogenous tumor antigens in the strong antitumor effects of GPC1-specific CAR-T cell therapy in the syngeneic tumor modes (Figure 4K).”

Discussion: “We demonstrated that CAR-T cells enhanced the CTL induction and generated immunological memory against non-GPC1 endogenous tumor antigens.”

Materials and methods: “For the re-challenge model, 5×105 parental MCA205 cells were subcutaneously inoculated into opposite site of the flank in naive mice with no history of bearing tumors or the mice rejected MCA-205-mGPC1 by the mCAR-T cell therapy. Parental MCA205 was inoculated 120 days after the mCAR-T administration.”

Figure 4 legend: “K) mGPC1-negative parental MCA205 was inoculated in naive mice with no history of bearing tumors and mice rejected MCA205-mGPC1 by mCAR-T cells injection 120 days after the mCAR-T cell administration. Tumor-growth curves of the individual mice in each group are shown.”

We also added the sentences explaining possible reasons for the discrepancy of the antitumor effects between the xenogeneic and syngeneic mouse model as follows:

Discussion: “These findings might explain stronger antitumor effects of mCAR-T cells in the immunocompetent syngeneic mouse models compared to the delayed antitumor effects of hCAR-T cells in the immunodeficient xenogeneic mouse model.”

4) Expression level of GPC1 in LK2-hGPC1 shown in Figure 2B and that in Figure 4B looks different. This information is important to know how high GPC1 expression in the mouse tumor cells in which GPC1 is enforcedly expressed.

The differences were caused by differences of flow cytometer settings. We reanalyzed GPC1 expression on theses cell lines under the same flow cytometer settings in the same experiment. We changed Figure 2 and Figure 4B and added the sentences in the Figure 4B legend as follows:

“B) The mGPC1-overexpressing murine cells, (MC38-mGPC1 and MCA205-mGPC1), endogenous hGPC1-expressing human cells (TE14), …”

Reviewer #2:In this study, the authors demonstrated that GPC1 could be a potential target of CAR-T cell therapy in some types of solid tumors. It was also shown that the anti-tumor efficacy of the anti-GPC1 CAR-T cells was augmented by the combination with anti-PD-1 antibody. Based on cross-reactivity of the anti-GPC1 Ab, they also proposed the safety of anti-GPC1 CAR-T cell therapy. While CAR-T cell therapy against solid tumors are of highly interest as the next generation of cancer immunotherapy, this study contains significant defects in the experimental data and conclusion led by the authors. Inclusion of additional data and discussions according to below comments are necessary.1) The authors concluded the safety of anti-GPC1 CAR-T cell therapy based on the cross-reactivity of Ab to mouse GPC1 and the similarity of GPC1 expression patterns between mouse and human normal tissues. However, this conclusion is hardly acceptable due to following reasons: 1) Even though their anti-GPC1 Ab cross-reacts with mouse GPC1, detail characteristics of the protein interaction are not necessarily identical. For example, affinity, avidity, Kd, and Ka of the Ab to human GPC1 and mouse GPC1 might not be the same. Moreover, the CAR construct utilized scFv instead of full form of Ab. Thus, detail characteristics as above must be assessed by using scFv. 2) As anti-GPC1 Ab used in this study was generated in chicken, immunogenicity of anti-GPC1 scFv will not be identical in human and mouse. Due to these reasons, observations in this study are not enough to conclude the safety of anti-GPC1 CAR-T cell therapy in human.

To evaluate detailed characteristics of the protein interaction between anti-GPC1 scFv and human or murine GPC-1 protein, we performed surface plasmon resonance (SPR) analysis using anti-GPC1 scFv and human and mouse recombinant GPC1 protein. The SPR analysis showed high binding affinity of anti-GPC1 scFv against both human and mouse GPC1 proteins as shown in Author response tables 1 and 2 below. Although the anti-GPC1 scFv shows higher affinity against hGPC-1 than against mGPC-1, its difference is less than 10-times and both affinities are higher than K_D_ 1x 10^7-8^, indicating that biological activities of anti-GPC1 scFv could be observed against both human and mouse GPC1. In in vitro study, both HL and LH versions of GPC1-specific CAR-T cells had antitumor responses to human and mouse GPC1 positive cancer cells without response to low GPC1 expressing human LK2 and murine MC38 cancer cell lines. These observations indicate that the GPC1-specific CAR-T cells may be utilized for the evaluation of antitumor effects and adverse effects in both human and mouse tumor models.

**Author response table 1. resptable1:** Anti-GPC1 scFv HL form.

	Ka (1/Ms)	Kd (1/s)	K_D_ (M)
hGPC1	1.40E+05	0.0017	1.22E-08
mGPC1	2.73E+04	0.002963	1.09E-07

**Author response table 2. resptable2:** Anti-GPC1 scFv LH form.

	Ka (1/Ms)	Kd (1/s)	K_D_ (M)
hGPC1	1.45E+05	0.00131	9.06E-09
mGPC1	4.21E+04	0.002179	5.18E-08

With these new findings, we added the sentences in the Results and Materials and methods sections as follows:

Results: “Surface plasmon resonance (SPR) analysis showed high binding affinity of LH or HL forms of scFv against recombinant hGPC1 protein as calculated K_D_ value 9.06 x 10^-9^ M or 1.22 x 10^-8^ M, respectively, which was as high as that of anti-CD19 scFv currently used in clinical settings (Ghorashian et al., 2019

Results: “To generate GPC1-specific CAR-T cells from murine T cells and evaluate antitumor effects and adverse effects on normal tissues in syngeneic mouse models, we evaluated binding affinity of both LH and HL forms of scFv against recombinant mGPC1 protein by SPR analysis, and found that both LH (K_D_ 5.18 x 10^-8^ M) and HL (K_D_ 1.09 x 10^-7^ M) forms of scFv showed high binding affinity against mGPC1, although less affinity than those against hGPC1. The HL form was used for generating GPC1-specific mCAR-T cells, because the HL form of hCAR-T cells showed higher antitumor activity in vitro (Figure 2C and D)”

Materials and methods: “Surface plasmon resonance (SPR) analysis

The sequences of anti GPC-1 scFv were cloned into pCAG-Neo mIgG2a-Fc plasmid (FUJIFILM Wako, Osaka, Japan) for the generation of anti GPC-1_scFv-mIgG2a_Fc fusion protein. […] The sensor chip was regenerated by 10 mM glycine–HCl (pH 1.7) for 0.5 min. Binding to the sensor chip is given as resonance units (RU). Data were analyzed by Biacore T200 Evaluation Software (v2.0) (GE Healthcare Life Sciences).**”**

The SPR analysis was performed by additional researchers “Haruhiko Kamada, Hiroki Akiba, and Kouhei Tsumoto”, so we added them as co-authors. In addition, some of the new experiments for revision of this manuscript were performed by Yuki Katoh and we changed his order in co-authors.

In terms of potential problems of immunogenicity of chicken based CAR-T cells in patients, it is difficult to speculate at this moment, unless performing clinical trials. However, non-humanized CD19-specific CAR-T cells have demonstrated significant antitumor effects for a variety of B cell malignancy possibly due to their relatively immunosuppressed condition via malignancy and chemotherapy / preconditioning. And strong antitumor effects of the chicken based GPC1 specific CAR-T cells showed in this study. Therefore, even this form of CAR-T cells could be used for clinical trials, although we are also considering further humanization for clinical trials. We added sentences in the Discussion as follows:

“Immunogenicity of the chicken based CAR-T cells may cause rejection of the CAR-T cells and poor responses in patients. […] Therefore, this form of CAR-T cells could be used for clinical trials, although we are also considering further humanization for clinical trials.”

2) As for anti-tumor effects of anti-GPC1 CAR-T cells, experimental models are not appropriate to evaluate its potency. In syngeneic mouse models, CAR-T cells were injected 3 days after tumor inoculation. Day 3 tumor size is too small to consider as pre-established tumor models. Experiments to treat late-stage tumors with larger size are necessary. In addition, 50,000 IU IL-2 was injected up to 6 doses after CAR-T cell therapy, which is unusual in mouse CAR-T experiments and will not be used in clinical settings. Anti-tumor efficacy of anti-GPC1 CAR-T cells without IL-2 injections are needed.

We did additional experiment to evaluatein vivo antitumor activity of our GPC1-specific CAR-T cells using large tumor model (tumor volume, more than 100 mm^3^ on day 7) and observed a significant antitumor activity without any obvious adverse effects. We added new Figure 4G and sentences in the Results and Figure 4G legend sections as follows:

Results: “The mCAR-T cells were also injected into mice bearing large MCA205-mGPC1 tumor (tumor volume > 100 mm^3^) and significant in vivo antitumor activity was observed without any obvious adverse effects (Figure 4G).”

Results: “Furthermore, administration of more than 10 times higher numbers of CAR-T cells did not show obvious adverse effects (data not shown) with significant antitumor effects against larger tumor (Figure 4G).”

Figure 4G legend: “G) Mice bearing MCA205-mGPC1 large tumor (tumor volume is > 100 mm^3^) received 3.5 x 10^7^ cells of mCAR-T cells or mCont-T cells on day 7 of tumor transplantation.Mean tumor volumes (mm3 ± SD) of each group (left panels) and tumor-growth curves of the individual mice in each group (right panels) are shown.”

As the reviewer pointed out, IL-2 administration was not used in current CD19-specific CAR-T cell therapies against relatively immunosuppressed patients with B-cell malignancies. However, the need of IL-2 administration in CAR-T cell therapies is still controversial in solid cancers. We decided to inject IL-2 following the report showing that more than 6 times injections of IL-2 effectively enhanced antitumor effects of the adoptive T cell therapies in syngeneic mouse models (Klebanoff CA et al. Clin. Can. Res. 2011.). IL-2 has previously been administered with the gene engineered T cell therapies, including CAR-T cell therapies especially for solid cancers (e.g. In mouse models; Chinnasamy et al.,2010., Tran et al., 2013., In human clinical trials; Robbins PF et al. J. clin. Oncol. 2011, Goff SL et al. J immunother.2019.).

3) Epitope spreading following anti-GPC1 CAR-T therapy was suggested based on IFN-g responses to gp70 in the treated mice (Figure 4I). However, it remains unclear whether the epitope spreading plays a crucial role in the anti-tumor effects of anti-GPC1 CAR-T therapy or is merely a by-stander phenomenon. In order to address this important question, efficacy of anti-GPC1 CAR-T cells should be examined in the mice with a depletion (or deficient) of endogenous T cell (or CD8 T cells). In addition, experiments to re-challenge GPC1-negative parental tumor into the mice which had rejected GPC1-positive tumor by CAR-T therapy are important.

To evaluate the importance of endogenous T cell responses to endogenous tumor antigens in the GPC1-specific CAR-T cell therapy, we performed additional experiments as suggested by the reviewer; re-challenge of GPC1-negative parental MCA205 tumor cells in the mice which had complete regression of MCA205-mGPC1 by the GPC1-specific mCAR-T cell therapy. The cured mice showed significant growth inhibition of the re-challenged tumors, indicating the strong endogenous tumor specific T cells which may be involved in the stronger antitumor effects of CAR-T cell therapy in syngeneic murine tumor models than those in xenogeneic mouse models. We added the new figure in Figure 4K and sentences in the Results, Discussion, Materials and methods and figure legend sections as follows:

Results: “Moreover, to evaluate the presence and importance of endogenous T cell responses in vivo, mGPC1-negative parental MCA205 tumor cells were re-challenged into the mice which had complete tumor regression by mCAR-T cell therapy in the MCA205-mGPC1 model (Figure 4F). These mice showed significant growth inhibition of the inoculated mGPC1-negative tumors indicating the induction and involvement of T cells specific for endogenous tumor antigens in the strong anti-tumor effects of GPC1-specific CAR-T cell therapy in the syngeneic tumor modes (Figure 4K).”

Discussion: “We demonstrated that CAR-T cells enhanced the CTL induction and generated immunological memory against non-GPC1 endogenous tumor antigens.”

Materials and methods: “For the re-challenge model, 5×105 parental MCA205 cells were subcutaneously inoculated into opposite site of the flank in naive mice with no history of bearing tumors or the mice rejected MCA-205-mGPC1 by the mCAR-T cell therapy. Parental MCA205 was inoculated 120 days after the mCAR-T administration.”

Figure 4K legend: “K) mGPC1-negative parental MCA205 was inoculated in the naive mice with no history of bearing tumors and mice rejected MCA205-mGPC1 by mCAR-T cells injection 120 days after the mCAR-T cell administration. Tumor-growth curves of the individual mice in each group are shown.”

We also added the sentences explaining possible involvement of endogenous antitumor T cells in the GPC1-specific CAR-T cell therapy as follows:

Discussion: “These findings might explain stronger antitumor effects of mCAR-T cells in the immunocompetent syngeneic mouse models compared to the delayed antitumor effects of hCAR-T cells in the immunodeficient xenogeneic mouse model.”

4) In NOG mouse model, CAR-T cells were injected 9 days after tumor inoculation, but the suppression of tumor growth was visible only after day 30 (Figure 2E). Please explain any reason why there is such a gap between CAR-T injection and the detection of anti-tumor efficacy. Were any GVHD symptoms such as body weight loss observed in these NOG mice with CAR-T treatment?

Delayed antitumor effects observed in the xenogeneic models may be explained by insufficient activation and expansion of hCAR-T cells in our xenogeneic mouse model condition. GVHD as suggested by body weight loss might help activation and expansion of hCAR-T cells and showed delayed antitumor effects in a GPC1 specific manner. We have previously reported that GVHD promote proliferation of administered human T cells in NOG mice (Tomonori Y et al. Cell Mol. Immunol. 15(11), 953-962, 2018). We added the sentences as follows

Discussion: “For example, delayed antitumor effects observed in the NOG mouse model (Figure 2E) compared to syngeneic mouse models (Figure 4E, F, and G) might be explained by xenogeneic GVHD reaction that might promote late activation and expansion of hCAR-T cells.”

5) MC38 is known as PD-1 Ab treatment-sensitive tumor. In this study, however, control T + anti-PD-1 Ab did not show any significant anti-tumor effects compared to control T + isotype Ab (Figure 6). Potential reasons for this observation should be explained.

One of the reasons for the relative resistance of anti-PD-1 Ab monotherapy in our mCAR-T cell/MC38-mGPC1 models may be explained by immunosuppression through total body irradiation performed as a preconditioning for in vivo mCAR-T cell expansion. Therefore, we added the sentences in the Results section as follows:

“Antitumor effects of anti-PD-1 Ab were relatively weak due to preconditioning with total body irradiation for in vivo mCAR-T cell expansion.”